# Towards guidelines to harmonize textural features in PET: Haralick textural features vary with image noise, but exposure-invariant domains enable comparable PET radiomics

George Amadeus Prenosil[1]*, Thilo Weitzel[1], Markus Fürstner[1], Michael Hentschel[1], Thomas Krause[1], Paul Cumming[1,2], Axel Rominger[1], Bernd Klaeser[1,3]

1 Department of Nuclear Medicine, Inselspital, Bern University Hospital, University of Bern, Bern, Switzerland, 2 School of Psychology and Counselling and IHBI, Queensland University of Technology, Brisbane, Australia, 3 Department of Radiology and Nuclear Medicine, Cantonal Hospital Winterthur, Winterthur, Switzerland

* george.prenosil@insel.ch

## Abstract

### Purpose

Image texture is increasingly used to discriminate tissues and lesions in PET/CT. For quantification or in computer-aided diagnosis, textural feature analysis must produce robust and comparable values. Because statistical feature values depend on image count statistics, we investigated in depth the stability of Haralick features values as functions of acquisition duration, and for common image resolutions and reconstructions.

### Methods

A homogeneous cylindrical phantom containing 9.6 kBq/ml Ge-68 was repeatedly imaged on a Siemens Biograph mCT, with acquisition durations ranging from three seconds to three hours. Images with 1.5, 2, and 4 mm isometrically spaced voxels were reconstructed with filtered back-projection (FBP), ordered subset expectation maximization (OSEM), and the Siemens TrueX algorithm. We analysed Haralick features derived from differently quantized (3 to 8-bit) grey level co-occurrence matrices (GLCMs) as functions of exposure $E$, which we defined as the product of activity concentration in a volume of interest (VOI) and acquisition duration. The VOI was a 50 mm wide cube at the centre of the phantom. Feature stability was defined for $df/dE \to 0$.

### Results

The most stable feature values occurred in low resolution FBPs, whereas some feature values from 1.5 mm TrueX reconstructions ranged over two orders of magnitude. Within the same reconstructions, most feature value-exposure curves reached stable plateaus at similar exposures, regardless of GLCM quantization. With 8-bit GLCM, median time to stability was 16 s and 22 s for FBPs, 18 s and 125 s for OSEM, and 23 s, 45 s, and 76 s for PSF

**Data Availability Statement:** All original PET/CT measurments are available from the osf database

under the link https://osf.io/36c7s/?view_only=
b7c9529129c649c3a32d5527df793265

**Funding:** The authors received no specific funding
for this work.

**Competing interests:** The authors have declared
that no competing interests exist.

reconstructions, with longer durations for higher resolutions. Stable exposures coincided in OSEM and TrueX reconstructions with image noise distributions converging to a Gaussian. In FBP, the occurrence of stable values coincided the disappearance of negatives image values in the VOI.

## Conclusions

Haralick feature values depend strongly on exposure, but invariance exists within defined domains of exposure. Here, we present an easily replicable procedure to identify said stable exposure domains, where image noise does not substantially add to textural feature values. Only by imaging at predetermined feature-invariant exposure levels and by adjusting exposure to expected activity concentrations, can textural features have a quantitative use in PET/CT. The necessary exposure levels are attainable by modern PET/CT systems in clinical routine.

## Introduction

Positron emission tomography in conjunction with computed tomography (PET-CT) is an imaging modality that quantifies radiotracer uptake in the living organism. With CT-based attenuation correction, and correction for scatter, randoms, and dead time, PET depicts absolute tracer uptake in units of concentration. In addition, the PET/CT system's spatially resolved measurements reveal textural features, yielding a metric for quantifying an imaged object's structure [1]. Textural feature analysis increases the information retrievable from PET/CT images, which is useful for characterizing and classifying tissue and lesions according to differences in morphology [2]. Adding texture analysis to a PET/CT readout can thus improve diagnostic sensitivity and specificity for certain tumor entities [3]. Notably, textural features are exploited as a surrogate measure of tissue metabolic heterogeneity [4], which can be an index of tumor malignancy, for example in the case of thymic epithelial cancer [5]. Alternately, changes in textural feature values depict response to treatment in esophageal cancer [6]. Expanding the feature space of PET/CT images makes texture analysis useful for pattern recognition techniques [7] and computer-assisted diagnosis (CAD).

Among the various mathematical methods to quantify image texture, partial gray level co-occurrence texture features are the best-established, having been introduced in 1973 by Haralick *et al.* [8] for the automated segmentation of satellite images. Sometimes also referred to as second order statistics, Haralick features have since found their way into radiomics [9] and subsequently PET/CT imaging [7]. However, images—and hence the measured tracer uptake values—typically differ between the various clinical PET/CT systems, even when imaging the very same object [10]. Distinct instrumentation and differences in image reconstruction algorithms impart characteristic imaging properties to PET/CT systems [11]. Resultant differences in imaging properties limit the comparability of datasets acquired on different PET/CT systems or with different acquisition protocols, and hamper the use of quantitative tracer uptake values in clinical routine or in multicenter clinical trials [12].

[13,14] Image noise, with its texture-like manifestations, varies especially with frame duration, tracer uptake, and the sensitivity of a given PET/CT system, which hampers comparison of datasets.

Despite certain efforts to quantify image noise and its effects on tracer uptake measurements [15–17] and image quality [18], the optimal image exposure remains unresolved for many clinical situations, and especially so in the context of textural feature values. These values must not only be stable in longitudinal, repeated measurements in a patient [19], but must be comparable between different PET/CT systems, if they are to be of diagnostic or prognostic value. The comparability problem of textural feature values is indeed analogous to the more widely recognized problem of variability in measured tracer uptake values.

Several works have examined textural feature stability [20], either under differing image acquisition and reconstruction conditions [21,22], in terms of grey level quantization [23], or under multicenter conditions [24], to identify the least variant features. However, the contribution of image count statistics, a known driver of variability in imaging, remains largely overlooked. Although, some studies show effects of noise on image texture [25–27], others have reported insignificant effects of image statistics on textural features [28,29]. In fact, there has hitherto be no systematic study of the progression of textural feature values with changing photon count. In most literature reports, the variability of textural features arising from image noise is usually described in terms of variance [30] and within the same exposure regime [31], without reporting on possible domains with stable feature values. This implies that radiomic values are not readily comparable between different PET/CT systems with different sensitivities and imaging properties.

The voxel (second order) neighborhood statistics of Haralick features must depend on their underlying first order statistics, i.e. the intensity distribution of the voxel image. At the detector level, Shot noise [32] will dominate in PET/CT and the signal-to-noise ratio (SNR) will be proportional to the square root of the imaged counts. Shot noise follows a Poisson distribution, but the noise distribution in the final PET/CT image will depend on the chosen image reconstruction method and its parameters. For examples, PET/CT images acquired with sufficient counts and reconstructed with filtered back projection (FBP) approximate Gaussian statistics [33], whereas ordered subset expectation maximization (OSEM) [34,35] or point spread function (PSF) based resolution modeling tend to resemble log-normal [36,37] or gamma distribution statistics [33]. Furthermore, with OSEM or PSF-based algorithms, the covariance matrix of a given image region will depend on the very region itself, e.g. the mean image [36]. This non-linearity renders mathematical predictive approaches to noise behavior for such reconstructions difficult and ill-posed. Image noise propagates linearly through the FBP reconstruction [38] or non-linearly, in an iteration dependent manner through OSEM- [36,39] and PSF-based reconstructions [40]. Therefore, noise increases the apparent roughness of the final image [17,41] differentially for different image reconstructions.

It follows that PET/CT acquisition and reconstruction protocols delivering consistent and comparable features values require prior knowledge of the dependency of textural features on the count statistics and on imaging properties. Only then can textural features become comparable across sites using diverse PET/CT systems, which is a necessary condition for quantitative usage of texture metrics in multi-center studies. Given this background, our aim was to investigate experimentally the count statistics dependency of first order statistical and Haralick features commonly used in PET/CT. For practical applications, we elaborate a methodology for identifying exposure domains with minimal variability in textural features. We apply our methodology on Haralick textural features measured in seven different image reconstructions, but our approach is generalizable to other image reconstructions and feature definitions.

To test feature stability, we obtained PET/CT measurements of a homogenous, solid-state Ge-68 phantom–just as used for daily quality assurance (QA) tests—in our clinical PET/CT system, with repeated measurements for durations ranging from three seconds to three hours. The product of acquisition times and phantom activity covered a wide exposure range. -We

measured the selected textural features within the same central region of the cylindrical phantom for seven clinical image reconstruction protocols, thus testing our hypothesis that textural feature values are highly variable depending on image noise, resolution, and reconstruction procedures. Additionally, we looked for criteria that convey stability to Haralick feature values in different image reconstructions. In so doing, we present a simple and automated way to identify exposures with stable image features for any given PET/CT system.

## Material and methods

### Data acquisition

Phantom measurements of varying exposure were acquired and reconstructed on a Biograph mCT-X 128 (Siemens Medical Solutions USA, Knoxville, TN) using the same Ge-68 solid-state phantom as used for daily quality assurance (QA) (Eckert & Ziegler Isotope Products, Valencia CA, USA). This cylindrical phantom had an inner diameter of 200 mm, cylinder wall thickness of 6.5 mm, and an active volume of 8407 ml of 68-Ge homogeneously distributed in a hardened epoxy matrix.

Our use of the QA phantom imparts an easily reproducible methodology in clinical settings. We axially aligned and centered the phantom in the PET/CT system's field of view (FOV) as specified for the daily QA, and imaged only a single bed position centered on the phantom. We acquired 20 PET/CT measurements with 25 different frame durations increasing from three seconds to a final 10861 seconds at a factor of $\sqrt{2}$ per frame. All data were reconstructed with CT-based attenuation correction, with scatter correction, and in the time-of-flight mode (TOF).

In the contexts of image noise and comparability, we prefer exposure to acquisition duration as a measure for the actual number of available decays. We define exposure as the product of activity concentration in a volume of interest (VOI) and acquisition duration or, in the case of relevant decay times, the integral of activity concentration over the acquisition time. In addition, by introducing a factor representing the relative sensitivity of another PET/CT system or e.g. differing decay branching ratios of other positron emitters, results can easily be rescaled to match differing systems and differing acquisition conditions.

### Image reconstruction

Seven different image reconstruction protocols combined three different image resolutions with three different reconstruction methods. All images were reconstructed with isotropic voxels and with post-reconstruction Gaussian filtering. The full width at half maximum (FWHM) of the Gaussian post-reconstruction filter ranged from 1 to 4 mm. Standard-resolution (SR) images had voxels of around 4 mm isometric spacing, whereas high-resolution (HR) images had 2 mm spaced voxels. Because (and contrary to FBP and OSEM) the Siemens TrueX algorithm enables higher resolution PET/CT, so called ultra-high resolution (UHR) voxels of around 1.6 mm isometric spacing were paired with said algorithm [42] in a single protocol. The three different reconstruction algorithms used were FBP, OSEM, and TrueX, the latter being a PSF-based resolution-modeling algorithm specific to Siemens PET/CT systems. Table 1 provides an overview of the seven different protocols and their exact parameters.

### Data analysis

A cuboid of around 50 mm edge length, which was centered in the daily QA phantom, defined the VOI from which all first and second order statistics were calculated (Fig 1A and 1B). The actual cuboid edge length in a given spatial direction was adjusted to the next even voxel

**Table 1. Parameters of image reconstruction protocol.**

| Name | Voxel Size (x * y * z; mm) | Voxel Volume (µl) | Algorithm | Iterations | Subsets | Gauss FWHM (mm) | Cuboid VOI ($n_x$*$n_y$*$n_z$ = N) |
|---|---|---|---|---|---|---|---|
| SR FBP | 4.07 * 4.07 * 4.0 | 66.3 | FBP | N/A | N/A | 4 | 12*12*12 = 1728 |
| SR OSEM | 4.07 * 4.07 * 4.0 | 66.3 | OSEM | 3 | 21 | 4 | 12*12*12 = 1728 |
| SR PSF | 4.07 * 4.07 * 4.0 | 66.3 | TrueX | 3 | 21 | 4 | 12*12*12 = 1728 |
| HR FBP | 2.03 * 2.03 * 2.0 | 8.29 | FBP | N/A | N/A | 2 | 24*24*26 = 14976 |
| HR OSEM | 2.03 * 2.03 * 2.0 | 8.29 | OSEM | 3 | 21 | 2 | 24*24*26 = 14976 |
| HR PSF | 2.03 * 2.03 * 2.0 | 8.29 | TrueX | 3 | 21 | 2 | 24*24*26 = 14976 |
| UHR Clinical | 1.59 * 1.59 * 1.5 | 3.79 | TrueX | 4 | 21 | 1 | 32*32*34 = 34816 |

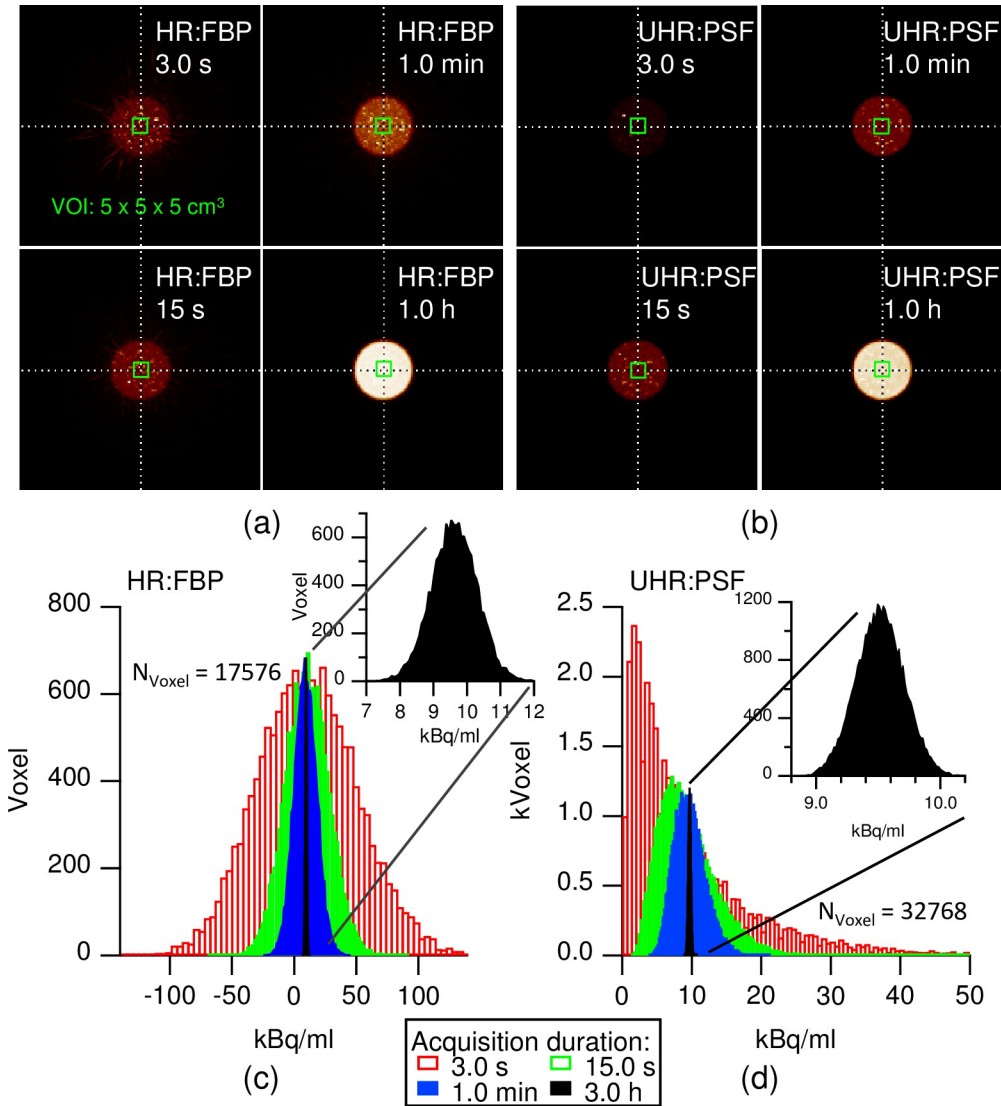

**Fig 1. Noise in PET/CT images at different exposures.** Shown are examples of four differently exposed HR FBP (a) and UHR PSF (b) images, using the daily QA phantom. Images are maximum intensity projections (MIPs) in the axial direction of the PET system with black set to naught and white set to the maximal intensity found in the PET image. The green rectangle marks the VOI, where all the calculations were performed. The associated noise histograms of the HR FBP (c) and UHR PSF (d) measurements are depicted below, showing the measured AC distribution in the VOI after four different acquisition durations. $N_{voxel}$ denotes the number of analyzed voxels in the VOI. Black insets are magnified histograms of the longest acquisition. Color legend of acquisition durations applies to both (c) and (d).

count, yielding arrays (c) with $n_x^* n_y^* n_z = N$ voxels. (cf. Table 1). Exposure $E_{Acq}$ in the cuboid VOI was defined as the product of the known activity concentration $AC$ in that region and acquisition duration, $t_{Acq}$

$$E_{Acq} = AC * t_{Acq} = [\text{nuclear disintegrations/unit volume} = 1/\text{ml}] \qquad 1$$

Our present use of exposure instead of acquisition duration for most graphs and measures makes results in this work independent of the actual AC, and readily transferable to other ACs. With 270.8 d half-life, the Ge-68 source retained a nearly constant activity concentration during the study, but decay would have to be considered in studies with short-lived nuclides.

The PET/CT data was automatically analyzed using a multi-paradigm software [43] written in-house. This software detected the phantom and calculated all statistics in a cuboid VOI placed automatically in the center of the phantom.

**Grey level co-occurrence matrices.** Six different grey level co-occurrence matrices (GLCMs) with 8 (3 bit), 16 (4 bit), 32 (5 bit), 256 (8 bit), 128 (7 bit) and 64 (6 bit) rows and columns were calculated from the central cuboid VOI, where each entry in the matrix represented the relative occurrence frequency $P(i,j) \equiv P(i,j|d = 1, \theta = 0° \cup 90°)$ of orthogonally adjacent voxel pairs with intensities $i$ and $j$. The number of rows and columns corresponded to G, the number of gray levels used. The relative occurrence frequency was calculated by dividing the number of $i,j$-voxel pairs ($p$) by the total number of voxel pairs $R$ found in the cuboid VOI.

$$P(i,j) = \frac{1}{R} \begin{bmatrix} p_{11} & \cdots & p_{1G} \\ \vdots & \ddots & \vdots \\ p_{G1} & \cdots & p_{GG} \end{bmatrix} \qquad 2$$

In this work, we used only orthogonal nearest neighbor voxels, with a displacement $d$ of one voxel and displacement angled $\theta$ of 0° and 90°. Diagonal voxels with $d > 1$ were excluded from consideration. We are aware of the GLCM's scale and direction sensitivity arising from use of different $d$ and $\theta$ values, but assuming isotropic image noise, no directional preference is to be expected. Furthermore, by using solely a nearest neighbor geometry and not a fixed distance $d$, the spatial sensitivity of the GLCMs scaled with the spatial sampling density of the PET/CT system and the chosen image resolution. The large number of voxels analyzed in the cuboid VOIs (cf. Table 1) justified the use of up to 256 gray levels, without producing information-free voids in GLCMs. This in turn enabled the statistical analysis of textural features using Haralick's methods.

*Grey level mapping in* GLCMs was performed as described in Tixier *et al.* [6]. This method resampled the gray levels found in a VOI to the finite range of the GLCMs, linearly mapping the lowest intensity in the VOI to the first gray level and the highest value to the last gray level of a GLCM.

**First order statistics.** In addition to the cuboid's mean $\mu_c$ and standard deviation $\sigma_c$, nine voxel value statistics were calculated from the central cuboid array $c$, as summarized in Table 2. These additional nine metrics, here called first-order statistics, often find use along with texture as descriptors of PET/CT images. In this work, we used Eqs 4–8 and 11 as descriptors for normality of the AC distribution in the VOI.

**Haralick features (second order statistics).** Eleven different Haralick features were calculated from the GLCMs as first described in Haralick *et al.* [8] and later summarized by Albregtsen [45]. Given below are the mathematical definitions of these texture features (Table 3).

**Table 2. First order statistical features used for texture analysis.**

| Feature Name | Equation | # |
|---|---|---|
| Coefficient of variation, CV | $= \frac{\sigma_c}{\mu_c}$ | 3 |
| Excess kurtosis | $= \frac{1}{N}\sum_{l=1}^{N}\left(\frac{c_l - \mu_c}{\sigma_c}\right)^4 - 3$ | 4 |
| Skewness | $= \frac{1}{N}\sum_{l=1}^{N}\left(\frac{c_l - \mu_c}{\sigma_c}\right)^3$ | 5 |
| Lilliefor's $D$ | The test statistic from a Kolmogorov-Smirnov test comparing the VOI value cumulative distribution $M_c$ to a cumulative Gaussian $G_0$ with same $\mu_c$ and $\sigma_c$: $D = \|M_c - G_0\|$ | 6 |
| Lilliefor's $p$ | Probability, that an observed $AC$ distribution conforms to a Gaussian calculated from the test statistic $D$ [44]. | 7 |
| Gauss Fit NRMSD | The normalized room mean squared deviation (NRMSD) of a Gaussian fitted to the $AC$ distribution in the VOI | 8 |
| Crest factor, CF (dB) | $CF = 20\log_{10}\frac{|c_{max}|}{RMS(c)}, \quad RMS(c) = \sqrt{\frac{1}{N}\sum c^2}$ | 9 |
| Efficiency, normalized entropy, $E$ | $E = \log_N\left(\prod_{i=1}^{N}P(i)^{-P(i)}\right)$, $P(i)$ = relative frequency of voxels with gray level $i$ in $\mathbf{c}$. | 10 |
| Mode / Mean | Ratio of the mode and the mean activity concentration found in the VOI | 11 |

**Table 3. Haralick features used for texture analysis.**

| Feature Name | Equation | # |
|---|---|---|
| Angular Second Moment (ASM), Homogeneity | $= \sum_{i=1}^{G}\sum_{j=1}^{G}\{P(i,j)\}^2$ | 12 |
| Contrast (CON) | $= \sum_{n=1}^{G}n^2\{\sum_{i=1}^{G}\sum_{j=1}^{G}\{P(i,j)\}, |i-j| = n$ | 13 |
| Inverse Difference Moment (IDM) | $= \sum_{i=1}^{G}\sum_{j=1}^{G}\frac{1}{1+(i-j)^2}P(i,j)$ | 14 |
| Entropy II (ENT) | $= -\sum_{i=1}^{G}\sum_{j=1}^{G}P(i,j) \times \log(P(i,j))$ | 15 |
| Correlation (CORR) | $= \sum_{i=1}^{G}\sum_{j=1}^{G}\frac{\{i\times j\}\times P(i,j)-\{\mu_x\times\mu_y\}}{\sigma_x\times\sigma_y}$ | 16 |
| Variance II (VAR) | $= \sum_{i=1}^{G}\sum_{j=1}^{G}(i-\mu)^2 P(i,j), \quad \mu = \frac{1}{G^2}\sum_{i=1}^{G}\sum_{j=1}^{G}\{P(i,j)\}$ | 17 |
| Sum Average (AVER) | $= \sum_{k=2}^{2G}k(P_{x+y})(k)$ | 18 |
| Sum Entropy (SENT) | $= -\sum_{k=2}^{2G}P_{x+y}(k)\log(P_{x+y}(k))$ | 19 |
| Difference Entropy (DENT) | $= -\sum_{k=0}^{G-1}P_{x-y}(k)\log(P_{x-y}(k))$ | 20 |
| Cluster Shade (SHADE) | $= \sum_{i=1}^{G}\sum_{j=1}^{G}\{i+j-\mu_x-\mu_y\}^3 \times P(i,j)$ | 21 |
| Cluster Prominence (PROM) | $= \sum_{i=1}^{G}\sum_{j=1}^{G}\{i+j-\mu_x-\mu_y\}^4 \times P(i,j)$ | 22 |

The mathematical terms and definitions used inside the feature equations are given below, starting with the mean in x- and y- direction of the GLCM matrix:

$$\mu_x = \sum_{i=1}^{G} i \sum_{j=1}^{G} P(i,j) \qquad\qquad 23$$

$$\mu_y = \sum_{i=1}^{G} \sum_{j=1}^{G} jP(i,j) \qquad\qquad 24$$

Standard deviation in x- and y-direction of the GLCM matrix:

$$\sigma_x = \sqrt{\sum_{i=1}^{G} \left(P_x(i) - \mu_x(i)\right)^2} \qquad\qquad 25$$

$$\sigma_y = \sqrt{\sum_{j=1}^{G} \left(P_y(j) - \mu_y(j)\right)^2} \qquad\qquad 26$$

Greyscale sum and difference vectors $P_{x+y}$ and $P_{x-y}$

$$P_{x+y}(k) = \sum_{i=1}^{G} \sum_{j=1}^{G} \{P(i,j)\}, i+j = k, for\, k = 2, 3, \ldots, 2 \qquad\qquad 27$$

$$P_{x-y}(k) = \sum_{i=1}^{G} \sum_{j=1}^{G} \{P(i,j)\}, |i-j| = k, for\, k = 0, 1, \ldots, (G-1) \qquad\qquad 28$$

Mode was calculated in x direction ($Mode_x$) as an additional statistical descriptor of GLCMs.

Feature values $F$ obtained from within the aforementioned VOI were plotted logarithmically against exposure $E$ and interpolated into continuous curves by locally estimated scatterplot smoothing (LOESS) curves using IGOR pro (Wavemetrics, Lake Oswego, Oregon 97035, USA), which was also used to generate graphs and tables. The LOESS was conducted with a sliding smoothing window with a width of thirteen measurement points. To equalize the contribution of geometrically increasing exposure values, we ran the LOESS on log-log data. The LOESS curves were then used to identify stable maxima and minima or plateaus, where the onset of a plateau is defined when the relative slope in LOESS curve became less than 5%/E, with E in units of kBq/ml*s.

$$relative\ slope = \frac{\Delta F}{\Delta E * F} \leq 0.05/E \qquad\qquad 29$$

## Results

### Noise distribution

To understand the impact of exposure upon image texture, PET/CT measurements of the daily GE-68 QA phantom were recorded at increasing acquisition durations. The phantom AC was 9.54 kBq/ml at the time of measurement, and thus the acquired exposures ranged from 28.6 kBq/ml*s to 103.6 MBq/ml*s with the chosen durations. Fig 1A illustrates how the apparent coarseness of the HR FBP images decreases with increasing acquisition duration. Fig 1B shows the same effect, albeit on a different resolution scale, for UHR PSF images. As examples, Fig 1C and 1D show the activity distribution histograms of the cuboid VOI in the cases of HR FBP and UHR PSF reconstructions. While activity histograms reconstructed with FBP tended be wider than were those from PSF reconstructions, the former were more Gaussian in

shape, even at brief acquisitions. In addition, FBP data displayed negative voxel values and required longer acquisition before their activity distributions moved away from zero than did OSEM (not shown) or PSF data. As expected, the spatial sampling density showed similar effects, with SR data having narrower and more Gaussian histograms than did HR or UHR data (not shown). The measured average activity concentration in the VOI was unaffected by exposure in all examined reconstructions (S1 Fig).

## Exposure dependency of first order statistical features

To reveal how noise propagates through an image reconstruction process, we calculated nine different statistical voxel value metrics from the central cuboids (Fig 2 and Eqs 3–11). When plotted against exposure, these measurements reflected the above findings in the histograms: While CV, crest factor, and normalized entropy followed the expected power law for all seven reconstruction protocols, excess kurtosis and skewness were constant for FBP but varied in OSEM and PSF data. In the latter datasets, skewness resembled CV with respect to exposure, but excess kurtosis initially followed a power law, only to stabilize around zero after exceeding some minimum exposure. SR data had no excess kurtosis, even at very low exposure values. The presence of negative values in FBP images as seen in Fig 1C caused a plateau to appear in the crest factor at low exposures. Negative values ceased to appear after 30 s in SR FBP and 679 s acquisition time in HR FBP data, translating to exposures of 286 kBq/ml*s and 6476 kBq/ml*s, respectively.

While kurtosis and skewness are accepted indicators for normality, we additionally used Lillifor's $p$ and the goodness of fit of a Gaussian to the noise distribution. Lillifor's $p$ remained close to unity for FBP data irrespective of exposure but increased from an initially low magnitude (10E-10) for OSEM and PSF data. Here, $p$ showed a sigmoidal approach towards a plateau at unity, in the same domain of exposure where excess kurtosis disappeared. Its associated test statistic $D$ is the supremum between the sample cumulative distribution and a normal distribution of equal mean and variance. Log plots revealed $D$ decreasing with exposure and following a power law, but with parallel ordinate shifts for the seven different reconstruction protocols. There was less pronounced exposure dependency of $D$ in FBP data, but the correlation was still visible.

Similarly, the Gauss fit NRMSD decreased with a power law with increased exposure for all seven reconstructions. As with the other metrics used, the AC distribution in FBP data more closely resembled a Gaussian distribution at lower exposures than required for OSEM and PSF data to attain a Gaussian. The mode/mean ratio reflected the increased skewness and non-Gaussian shape of the AC distribution at low exposures, while approaching unity in congruence with the plateauing of the Gauss fit NRMSD.

## Exposure dependency of grey level co-occurrence matrices

As a next step, we calculated GLCMs with different numbers of greyscales, mapping the PET data over a range extending between the respective minimum and maximum activity concentrations found in the cuboid VOI. Fig 3 shows examples of the GLCM dependency on acquisition duration for different HR reconstructions before greyscale mapping. Similar to the noise histograms noted above, these FBP GLCMs were broader, more Gaussian and less skewed than were OSEM or PSF GLCMs. The latter two types of GLCM showed an alignment on the diagonal at high exposures, with more skewness and some dispersion at low exposures. When shown in units of activity concentration, all GLCMs had exposure-invariant average values ($\mu_x$ and $\mu_y$; Eqs 23 and 24). By contrast, Mode$_x$ approached $\mu_x$ only at higher exposures, while $\sigma_x$ (Eq 25) showed the expected power law behavior.

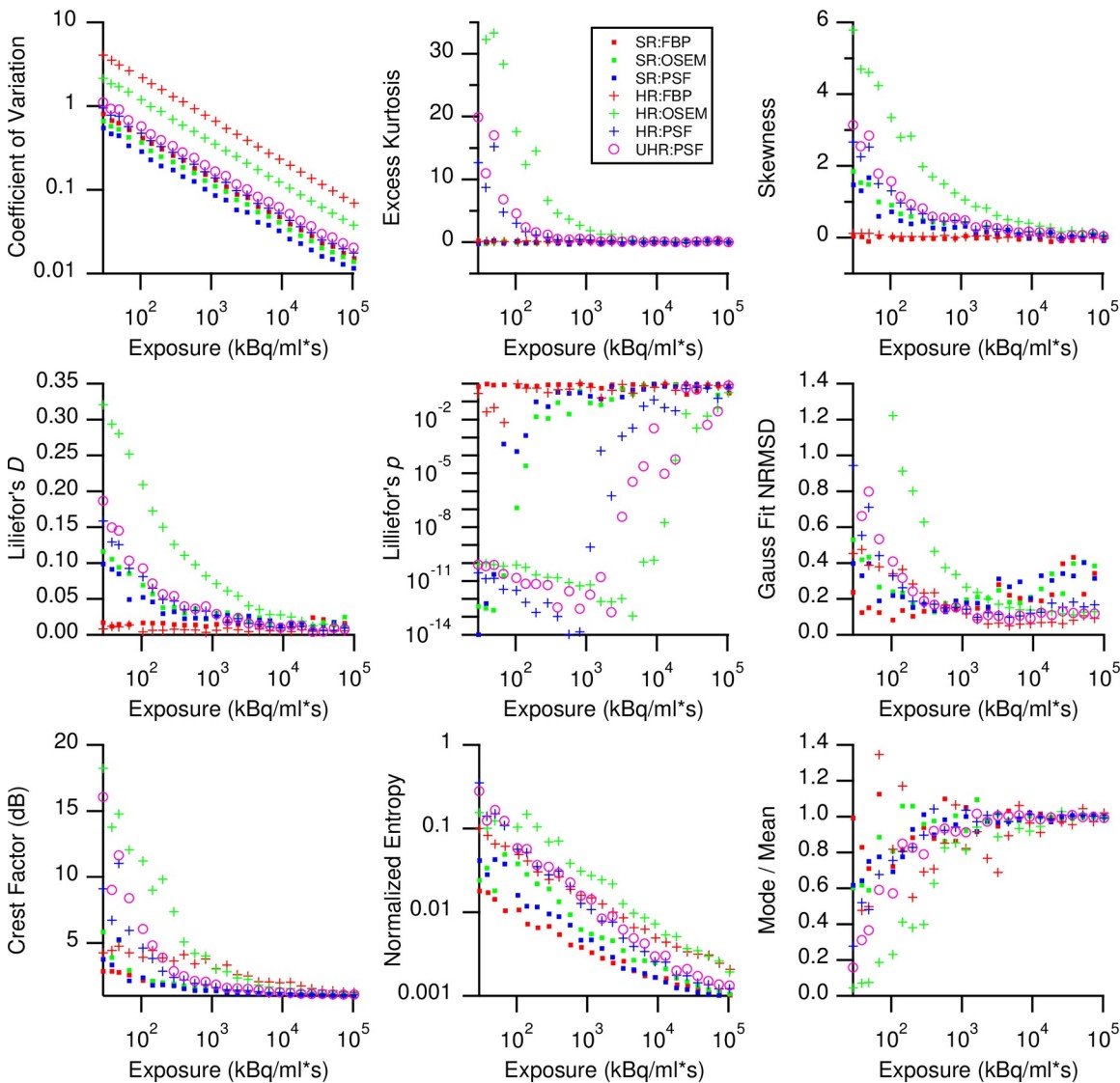

**Fig 2. First order statistics of image noise measurements as a function of exposure and grouped according to different reconstruction protocols.**

After greyscale mapping, the shape of GLCMs remained, but only GLCMs from FBP had a constant $\mu_x$, Mode$_x$, and $\sigma_x$. In OSEM and PSF reconstructions, these values showed a sigmoid function (Fig 4). Comparing Figs 2 and 3 suggests that greyscale mapping produced GLCMs of similar extension with regard to the grey scale count, because only using values occurring inside the VOI. Especially in the case of FBP, this led to GLCMs with a Gaussian cross-section of similar extent, while PSF based reconstructions still had GLCMs of varying shape and extent. Mapping the GLCMs in greyscale units maintained the general shape of the matrices, but revealed how $\mu_x$, Mode$_x$, and $\sigma_x$ moved towards higher values with increasing exposure in OSEM and PSF reconstructions, while remaining constant in FBP reconstructions (Fig 4).

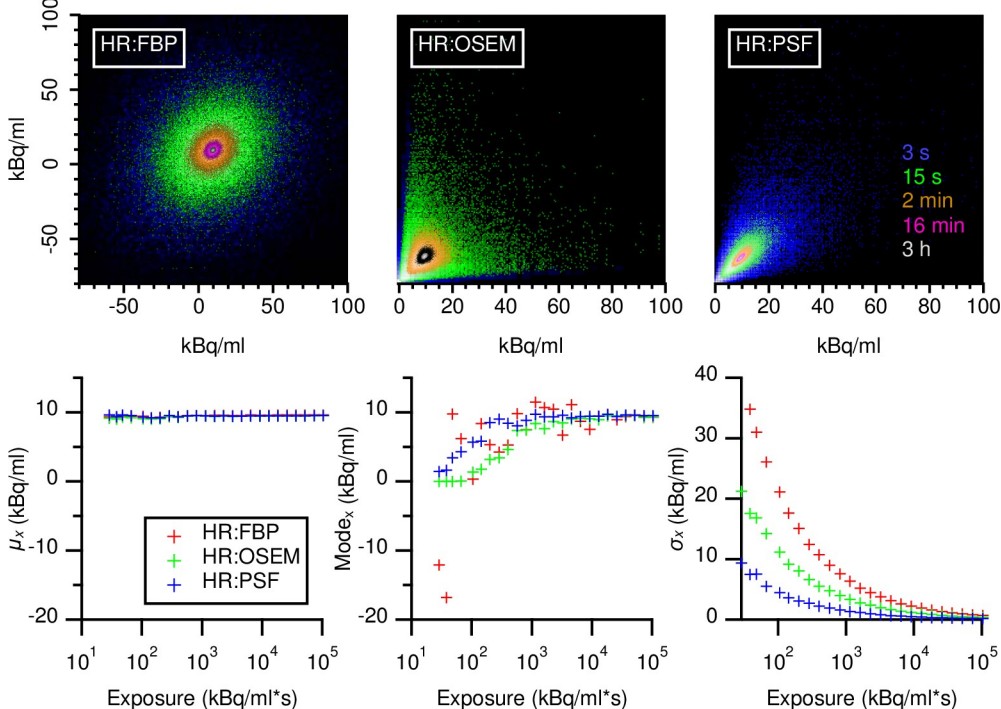

**Fig 3. Overlaid examples of 8 bit GLCMs in units of kBq/ml.** GLCMs calculated from HR datasets and reconstructed with either with FBP, OSEM, or TrueX, PSF. Five selected acquisition durations are shown at their location in respect to absolute activity concentrations. For visualization (top). GLCM maxima are normalized to unity and are displayed with a logarithmic color scale. Acquisition times are noted besides the PSF GLCM in the respective color.

## Exposure dependency of Haralick features

From the full range GLCMs, 11 Haralick or so called second-order statistical features were calculated and plotted against exposure (Eqs 12–22). The resulting scatterplots were smoothed by the LOESS method to obtain differentiable curves for every feature (Fig 5). Most feature-reconstruction combinations were variant at lower exposures before reaching an invariant plateau. Feature values from FBP stayed mostly constant over all exposures. The feature value curves behaved similarly in shape regardless of GLCM bit size. Nevertheless, varying the GLCM bin size resulted in differing absolute feature values; something that had been reported before [46].

Fig 6 shows the magnitude of variability between the lowest and highest values found in each textural feature calculated from the different GLCMs, as well as those found in first order statistics. While the highest variability of features extended over three orders of magnitude, the variability was almost similar between GLCMs regardless of their quantization levels (Figs 6 and S2A). Lowering the number of GLCMs greyscales led overall to a significant decrease in feature value range, but the effect size was negligible, with 0.013 to 0.028 Bel (S2B Fig).

We also observe that SR FBP generally provided for more stable features than did HR OSEM and HR PSF reconstructions. Among the individual features, ENT, CORR, SENT, and DENT were the most stable, whereas SHADE and PROM were the most unstable.

First order statistics, which are sometimes also used as image descriptors, varied up to three orders of magnitude with exposure (S1C Fig), with exception of Lilliefor's $p$, here used only as a Gaussian metric. Contrary to Haralick features, first order statistics mostly follow a power law concerning exposure and do not attain stable domains *per se* (Fig 2). Therefore, highest possible exposures should be sought when using first order statistics in PET/CT.

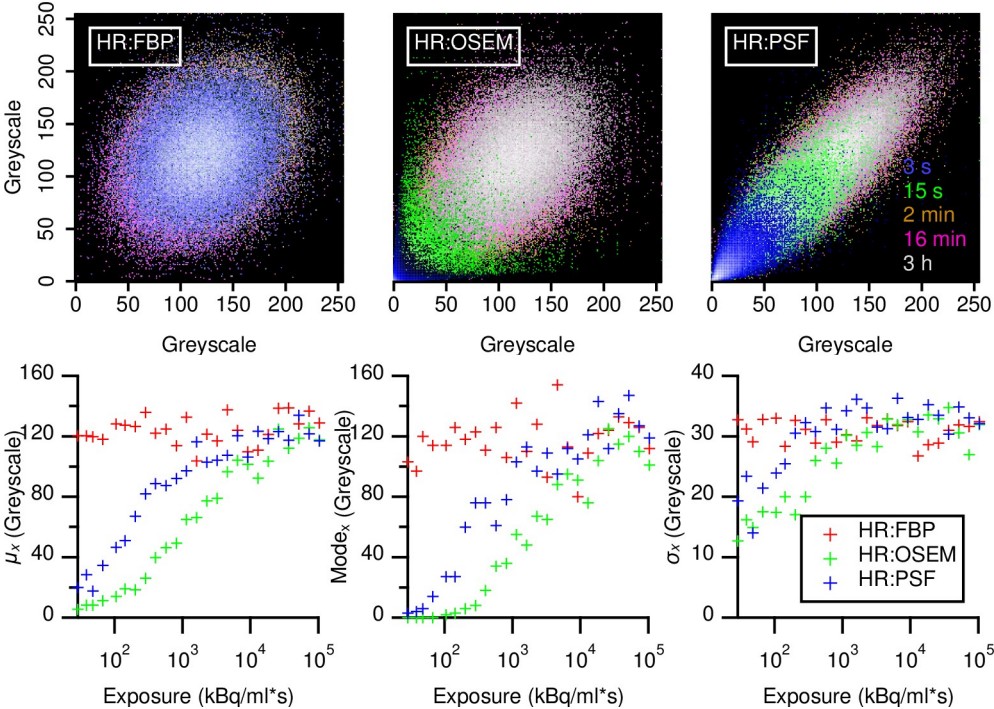

**Fig 4. Overlaid examples of 8 bit GLCMs in units of greyscales.** GLCMs calculated from HR datasets and reconstructed with either with FBP, OSEM, or TrueX, PSF. Five selected acquisition durations are shown at their location in respect to absolute their greyscale values. GLCM maxima are normalized to unity and are displayed with a logarithmic color scale. Acquisition times are noted besides the PSF GLCM in the respective color.

In SR reconstructions stable plateaus in Haralick features occurred at lower exposures than in HR reconstructions. Some feature-reconstruction combinations were stable over the entire exposure range (Marked with asterisks in Fig 7). Still, other feature-reconstruction combinations, such as SHADE in SR OSEM, attained stability distinctively later than the median exposure in their group. In terms of acquisition duration, median time to stability did not differ significantly between the differently sampled GLCMs (Fig 7). For example, when using 8-bit GLCM, median time to stability was 16 s for SR and 22 s for HR FBP, 18 s for SR and 125 s for HR OSEM, and 23 s, 45 s, and 76 s for SR, HR and UHR PSF reconstructions, respectively. In addition, we note that the occurrence of stable feature values in FBP reconstructions coincided the disappearance of negatives image values as occurs above certain exposures. In OSEM and PSF reconstructions stability coincided with the approach of noise histograms towards normality.

Because texture feature values also depend on the GLCM quantizing scheme [46], we studied the effects of differently quantizing GLCMs on feature stability. In these calculations, we used the full range of activity concentrations delivered by the PET/CT system for quantizing GLCMs (S3 Fig) rather than normalizing the grey level range to the intensities in the VOI. Without this greyscale mapping, long-term exposure stability failed to occur, although there emerged what we term "islands of stability" (S4 Fig). The resulting feature value-exposure curves were highly non-monotonic, and often had no distinct plateaus (S5 Fig). Variability was up to two orders of magnitude higher (S6A Fig). However, islands of stability occurred at similar exposures as the onsets of plateaus described earlier (S6B Fig). See Supporting Material and Methods (S1 Text) and Supporting Results (S2 Text) for details.

In cases where the greyscale mapping was restricted to a fixed bin size scheme with a lower bound on zero (c.f. S1 Text) as suggested by Presotto *et al.* [26], variability inflated to four

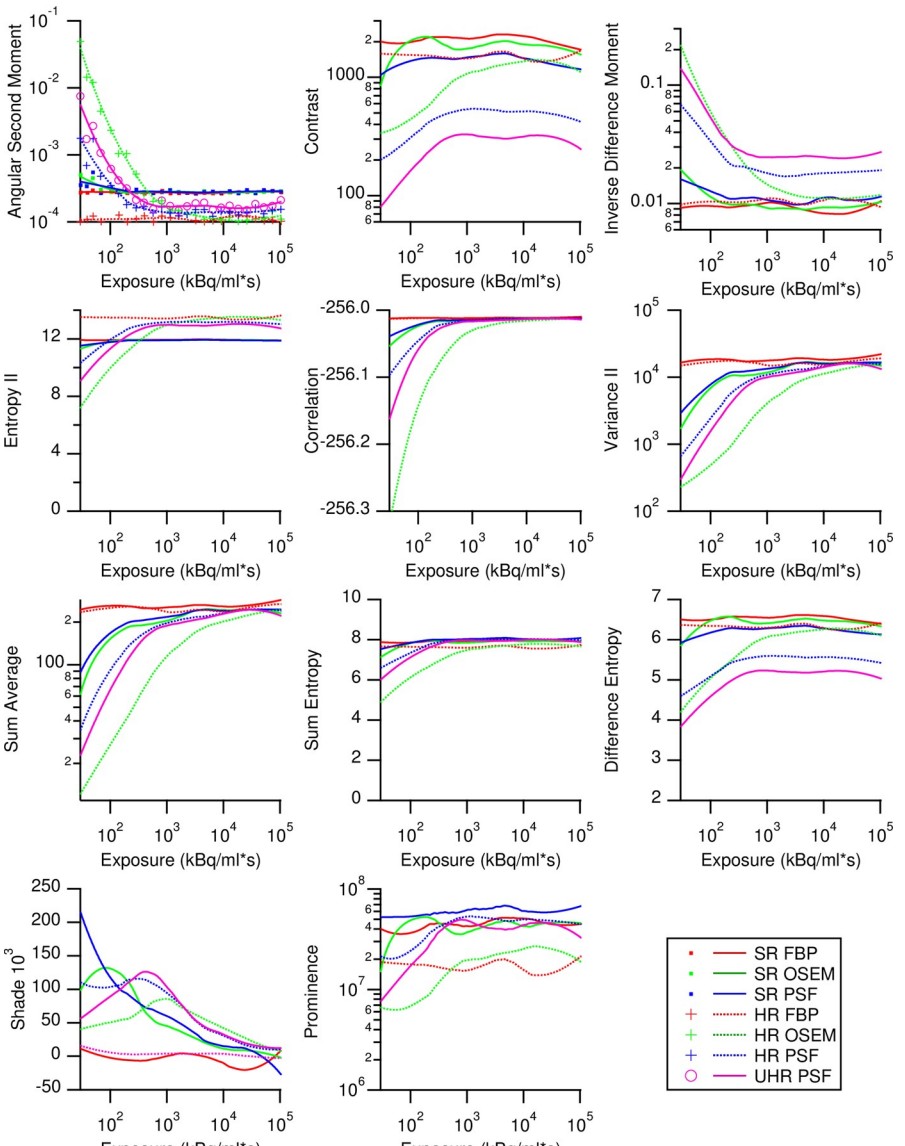

**Fig 5. Exposure dependency of Haralick texture features according to different image resolution and reconstruction algorithm calculated from 8-bit GLCMs with 256 grey levels.** Measurement points are shown in the first graph but omitted in the following ones for clarity, and only the interpolated Loess curves are shown. Loess confidence interval are not shown. In the ASM panel, a clinical relevant exposure range is shown in grey, taken from Krarup *et al.* [47]. Legend applies to all.

orders of magnitude, with almost no asymptotic plateaus evident in the feature value-exposure curves (S7 Fig).

## Discussion

This work shows how exposure and hence image noise affects Haralick texture features on PET/CT measurements obtained with the homogenous daily QA phantom. By varying exposure (Eq 1) over a wide range and including values relevant for clinical practice, we report that feature values varied by more than two orders of magnitude in PET/CT images of an otherwise absolutely homogeneous phantom. Additional variability arose due to the different image

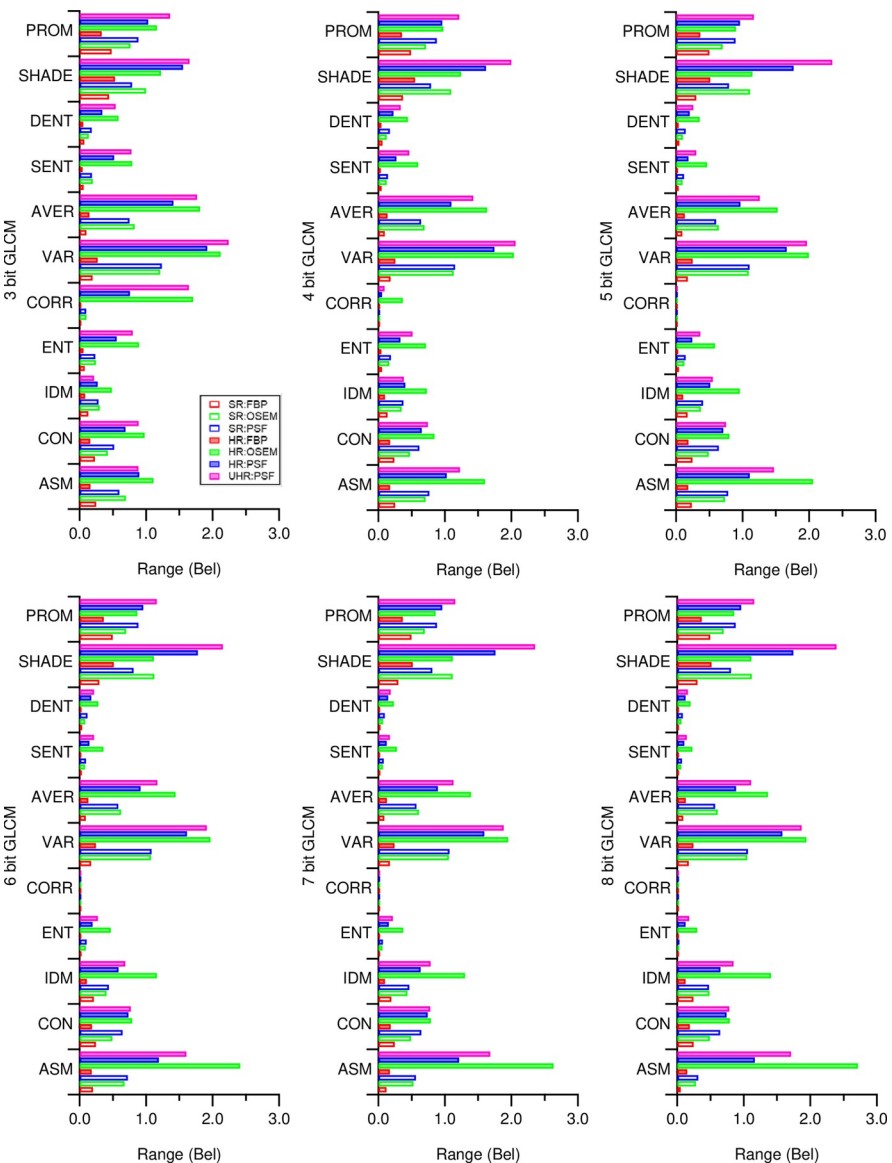

**Fig 6. Magnitudes of textural feature variability.** Shown is the absolute range between the lowest and highest values in units of Bel found in the texture features across all examined exposures calculated 8-bit GLCMs (a), from 7-bit GLCMs (b), from 6-bit GLCMs and were also found in first order statistics (d). Lilliefor's *p* varied up to 12 orders of magnitude and is therefore not shown to scale. Color Legend in (a) applies to all.

reconstruction methods, with FBP producing the most robust Haralick features and non-linear iterative image reconstructions (PSF and OSEM) producing the largest variability. Consequently, image resolution correlated inversely with the exposure levels needed for obtaining stable Haralick feature values. Most feature-reconstruction combinations showed monotonic curves with stable plateaus (Fig 5) when plotted against exposure. Greyscale mapping produced GLCMs with congruent positions within the matrix, making full use of the available matrix size (Fig 4). On the other hand, full-range GLCMs drifted across the matrix according to the grey scale range occurring in the image (S2 Fig). This additional degree of freedom could explain the non-monotonic curves and the larger variability of full range GLCMs (S4 and S5 Figs and S2 Text).

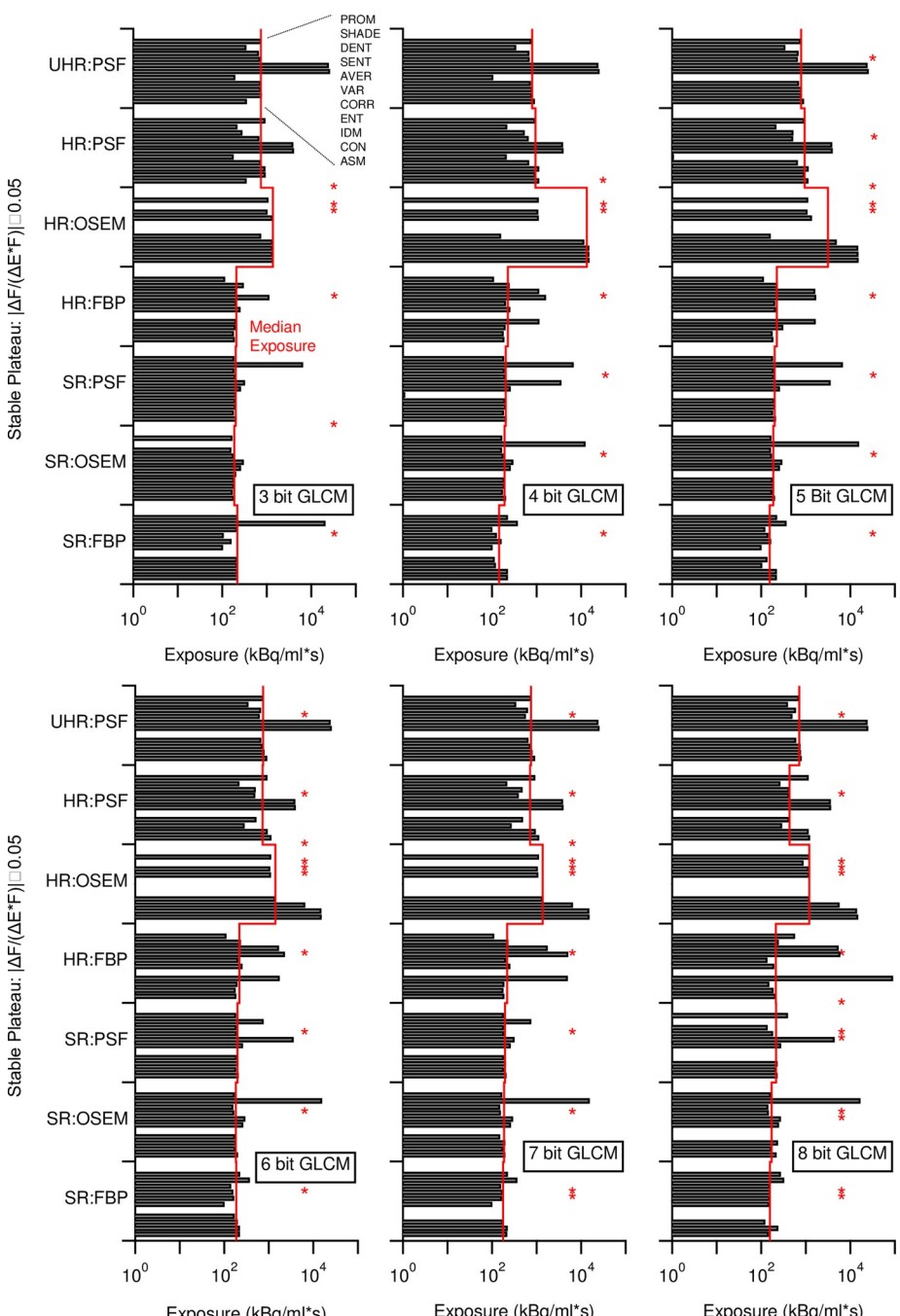

**Fig 7. Onsets of stable plateaus in features calculated from differently quantized GLCMs with median exposure as a red line.** *Missing values indicate stability over the entire exposure range. Textural feature order applies to all sets of image reconstructions.

The large variability in feature values is a function of how PET-CT image noise propagates into the final image: While FBP tended to produce Gaussian intensity distributions in homogenous VOIs in the phantom regardless of exposure, OSEM and TrueX reconstructions approached normality only with increasing photon count. The shape of these first order statistical distributions subsequently defined shapes of the (second order) GLCMs from which all

Haralick texture features were eventually calculated. Consequently, once the first order distributions attained normality, most of the feature values became exposure-invariant. In this respect, the median exposure for stable plateaus (Fig 7) coincided with the exposures needed to bring Lilliefor's *p* to unity (Fig 2).

Textural features in medical images occur as a function of the spatial distribution of the signal intensity in the structure. At the same time, this signal distribution is contaminated with noise. When signal is low within a given structure, the textural features may not emerge within a reasonable acquisition time, given the constraints of clinical practice. Moreover, regulations of radiation protection (diagnostic reference levels) limit the dose of injected tracer. These factors contribute to the difficulties in comparing texture features between sites.

The exposure regime where Haralick features become stable is within a clinically relevant range. Looking at the minimal mean uptake of 4 kBq/ml (calculated from $SUV_{mean}$ values and de-corrected for decay) encountered, for example, in lung cancer [47] and a median exposure of 729 kBq/ml for HR PSF reconstructions (6-bit GLCM, Fig 7), the minimal exposure to obtain more than 50% of exposure-invariant Haralick features is 182 s. This exceeds the 120 s used by the authors of the study [47], and furthermore, the features ASM, CON, VAR, and AVER remain influenced by exposure. From the minimal and maximal $SUV_{mean}$ in the study [47], we calculated and displayed an approximate clinical exposure range in the ASM panel of Fig 5, the strongest study correlate. One might ask, if better exposure control would have decreased the number of patients needed for the study or given better results to other Haralick features.

In dynamic clinical studies, acquisitions times for time activity curves are often very brief, and feature analysis is not regularly done. Nevertheless, let us assume a tracer uptake of around 8 kB/ml as seen in a single lesion (de-corrected for decay) in the work of Schwartz *et al.* [48]. Our results in Fig 7 then suggest minimal acquisition durations of 27 s and 89 s respectively would be required to obtain stable values on our PET/CT system. Such brief acquisition times bring feature analysis of dynamic acquisitions into the realm of the possible in a clinical setting, even though the minimal acquisition times still exceed the 10 s intervals used by Schwartz *et al* at the beginning of their time activity curves. As a matter of fact, with the somewhat lower sensitivity of their PET/CT system [49], the exposures from their 10 s intervals are approaching those from our briefest acquisition duration. Furthermore, in the case of UHR PSF reconstructions with 25 MBq/ml*s exposure required to obtain stable VAR values (Eq 17), the acquisition time soars to 52 minutes, assuming no increase in injected dose is sought.

Greyscale mapping, by giving stable plateaus (Fig 5) and lowering variability [24] (Fig 6), improves the stability and comparability of all feature values, making it a prerequisite for any feature analysis. Furthermore, stability occurs at lower exposures with greyscale mapping (Figs 7 and S5). However, this probably only holds true in a perfectly homogenous VOI, such as used in this work: Figs 4 and S5 and S7, and the work of Presotto *et al.* [26] hint that the quantization distance between grey level groups can introduce new variability. Such a situation might occur were lesions of similar texture are found within differing background ACs. It is therefore unsurprising that studies without greyscale mapped GLCMs report high feature variability [21], whereas studies using greyscale mapping yielded more reproducible results [19,24]. Furthermore, when using a fixed bin size scheme with the lower bound on zero [26], we found increased variability in the absence of exposures with stable plateaus (S7 Fig). However, neither feature variability (Fig 6) nor the occurrence of stable plateaus (Fig 7), were affected much by GLCM bit size.

We tested three types of standard image reconstruction algorithms, namely FBP, OSEM and TrueX at standard (4 mm) and high (2 mm) resolutions, and a seventh TrueX reconstruction at ultra-high resolution (1.6 mm). Through these systematic measurements, we found for

the SR dataset a large subset of features that proved to be stable at an exposure of approximately 200 kBq/ml*s. Excursions to high resolution yielded stable features only at 10-fold higher exposure, but low-resolution reconstructions required exposures falling in the range encountered in clinical routine. However, we also found 14 feature-reconstruction combinations with stability over the entire exposure range. On the other hand, angular second moment, variance, cluster shade, and prominence calculated from normalized GLCMs all showed exposure variability of up to 2.5 magnitudes when used in combination with HR OSEM and HR PSF reconstructions (Fig 6B). In a clinical setup, there is always a tradeoff between diagnostic utility of features and variability brought by the use of these features.

Fig 7 shows domains of stability occurring for most of the eleven features at standard resolution without a requirement for high or excessive exposure times. This finding is a matter of practical interest in clinical routine, and demonstrates the usefulness of feature values at our standard resolution. Because our objective is to obtain the best possible image texture information without increasing exposure, the chosen resolution has to match the clinical question. First and foremost, a chosen image resolution must support the visualization of a lesion's true texture to justify the use of texture analysis over a simple tracer uptake analysis. Next, the resolution must provide for sufficient voxels in a lesion to satisfy the statistical nature of Haralick features; small lesions might simply not have enough neighboring voxels for a concise feature description. Our results are not promising for the detection of features in lesions measuring < 12 mm, i.e. lesions with less than three SR-voxels in any direction.

Using a daily QA phantom for estimating correct acquisition times facilitates confirming that a given PET/CT system together with the chosen reconstruction method suffice to yield useful images within a practical time and tracer dose limits. The required acquisition duration will vary in conjunction with sensitivity of different PET/CT systems. All else being equal, a system with tenfold lower sensitivity than ours would require ten times longer scan durations for the same task. Furthermore, differing definitions of how GLCMs are calculated prevent the use of simple look-up tables for exposures, and thus mandate individual phantom measurements. With this work, we present an easily repeatable procedure to identify stable exposure domains, where image noise will not substantially add to textural feature values. These domains should be seen as minimal requirements for comparable radiomics in a multicenter setting. In addition, to quantify a heterogeneous texture, one must first be able to identify truly homogeneous areas, where any seeming texture is actually image noise.

However, there are certain caveats to using a perfectly homogeneous phantom. First, as the covariance matrix in OSEM and PSF based image reconstructions depends on the very object being imaged [36], making detailed predictions about the exposure dependence of texture feature values becomes difficult in heterogeneous clinical objects. Additionally, the non-Gaussian, object-dependent PSFs of said image reconstruction algorithms [50] violate the scale space axioms, stating that the imaging process should create no new image structures [51]. Particularly, PSF based reconstructions violate the axiom of non-creation and non-enhancement of local extrema and, and as we have shown previously [50], the scale invariance axiom. Furthermore, due to regularization processes in these algorithms, the membership of images created with PSF-based reconstructions in the Lie-group is disputable over the relevant scales. Therefore, acquisition durations determined for OSEM and PSF reconstructions depict only a general minimum time domain where proper texture analysis becomes possible. However, some of these problems could be compounded for inhomogeneous phantom measurements with real texture. Nevertheless, it is important to keep in mind that textural features calculated with nearest neighborhood operations by definition represent properties found at the highest spatial frequencies of the image. The highest spatial frequencies are precisely those most affected by the actual modulation transfer function (MTF) of the tomograph, by image resolution in

terms of sampling, by the SNR, and by non-Gaussian distributions of image noise. In fact, textural features calculated from differing sampling densities characterize different features found at different spatial frequencies. Such features are scale-variant. This state of affairs calls for the use of scale-invariant phantoms, because any scale-variant phantom will produce differing results for differing spatial resolutions. As such, it can be said that a homogenous phantom fits all spatial resolutions and reconstructions.

As a second caveat, the recovery of decays depends also on the depth of the VOI within the body and the size of the body [31]. Using the daily QA phantom, with its inner diameter of 200 mm, can thus lead to underestimated acquisition durations in the case of deep lesions with higher attenuation and scatter. Therefore, calculating the correct exposure for feature detection must consider the lesion depth and the applied scatter and attenuation correction. Our work used the deepest possible location for the chosen VOI; shallower locations can only bring an earlier onset of stable plateaus. Finally, our present use of the long-lived 68-Ge-radionuclide in the phantom does not emulate the decreasing signal to noise ratio occurring with the decay of short-lived clinical PET tracers. However, this last pitfall is easily correctable by using the expected (integral of) total decays instead of the decay-corrected activity concentration of the nuclide for dynamic exposure calculations. Nevertheless, for the case of [68-Ga]-PSMA-11 imaging, with its 68 min physical half-life, correct exposures may be unobtainable in low uptake lesions. When using different nuclides, their branching ratios must be considered to calculate exposure values. Because positron trajectories are de-correlated from the MTF, differences in positron range are irrelevant between isotopes. Additionally, the transconvolution method allows for recasting images acquired with different positron emitters [52].

Our findings stand in contrast to those of other works [28,29], where feature values were deemed noise-invariant. Besides clearly showing emergence of texture at low exposures (Fig 1A and 1B), our data shows noise-dependence of texture. Because texture stability occurs at sufficiently high exposures, variability is easily overlooked in studies with long acquisitions such as performed in [28,29]. Furthermore, features calculated from neighboring voxels ($d$ = 1voxel) describe texture exactly at the Nyquist frequency of an imaging system. Therefore, having a PSF with an effective FWHM of three to four times the voxel size [29], neighboring voxels will—regardless of the underlying texture or noise—show similar intensities. In this case, texture values will appear exposure-invariant, or the variance occurs at lower exposures. This apparent feature stability emerges because the MTF of such an imaging system will effectively filter structures and noise alike. Using a different feature definition with a displacement $d > 1$voxel would be advisable in cases where image oversampling is desired.

## Conclusion

Proper textural feature analysis is attainable only when respecting the limitations of the PET/CT system, and with appropriate adjustment of the acquisition duration. While differences in imaging properties, e.g. how the image is formed, can be overcome with transconvolution [10], the range of sensitivity of contemporary PET/CT systems mandates individual adjustment of exposure times. Otherwise, a comparative use of statistical texture features across different PET/CT devices or even differing reconstruction algorithms is destined to fail. In the context of multicenter studies, harmonized acquisition protocols should match spatial resolution and sampling, but should actually use non-standard acquisition durations. These non-standard acquisition durations require individual tailoring to a particular clinical site and question. In this respect, we present a methodology to perform measurements to arrive at minimum exposures required for robust feature value identification before even proceeding to

clinical imaging. Not only will the subsequent clinical measurements then vary less, but also expected feature values can be estimated for a particular PET/CT site.

Choosing a sub-set of stable features together with an appropriate image reconstruction protocol is every bit as important as adjusting exposure to fit the expected domain of stability. Based on our findings, we recommend using different image reconstructions for feature analyses (FBP or OSEM) and for viewing (PSF). When using FBP, exposure must be high enough to avoid negative image values. We also suggest always using greyscale mapping, together with grey level invariant feature descriptions [23]. An additional benefit of adjusting exposure for optimal texture detection will be lower sample size requirement in multicenter studies and, as is becoming ever more important, for machine learning algorithms in medical imaging. Furthermore, harmonizing acquisition protocols as suggested above could be the key to bringing texture feature analysis into clinical routine. Naturally, our findings from a homogeneous phantom require confirmation in the clinic for a wide range of tumor entities and tissue types. Furthermore, texture metrics other than Haralick features exist also merit close investigation.

## Supporting information

**S1 Fig. Variability of first order statistics.** Mean activity concentration $\mu_c$ in the VOI as a function of exposure and grouped according to different reconstruction protocols. (a) SR acquisitions, (b) HR and UHR acquisitions. (c) Shown is the absolute range between the lowest and highest values in units of Bel found in first order statistics. Liliefor's $p$ varied up to 12 orders of magnitude and is not shown to scale.
(PDF)

**S2 Fig. Effect of GLCM greyscale quantization on magnitudes of textural feature variability.** (a) Cumulative histograms of absolute ranges between the lowest and highest values in units of Bel found in all texture features from differently quantized GLCMs. (b) Histograms of differences in feature range from differently quantized GLCMs, i.e. effect size.
(PDF)

**S3 Fig. Full range GLCMs.** Overlaid examples of five full range GLCMs calculated from six differently reconstructed datasets with five selected acquisition durations. GLCM maxima are normalized to unity and are displayed with a logarithmic color scale. Acquisition times are noted adjacent to each SR FBP GLCM in the respective colors. $N_{Pairs}$ denotes the number of GLCM pairs analyzed for every color-coded matrix. Color scale in HR:FBP applies to all. Example GLCMs for HR:PSF are not shown, due to their similarity to the UHR:PSF measurements. Greyscales are truncated at 400.
(PDF)

**S4 Fig. Islands of stability.** Calculating islands of stability for the case of the inverse difference moment (IDM) feature according to the seven different reconstruction protocols, and for full range GLCMs. Plots depict the LOESS curves (colored solid lines) of the data (symbols) with their 99% confidence intervals (dotted lines). Black solid and dotted lines show the respective first derivatives of the Loess curves, with zero crossings at the intersections with the dashed horizontal line. Black crosses mark the islands of stability found for the IDM. The middle legend applies to all but the UHR PSF data.
(PDF)

**S5 Fig. Feature values calculated from full range GLCMs.** Exposure dependency of Haralick textural features according to different image resolution and reconstruction algorithm calculated from full range GLCMs with 256 grey levels. Measurement points are shown in the first

graph but omitted for clarity in subsequent. Loess curves are shown without associated confidence intervals.
(PDF)

**S6 Fig. Variability and stability in features from full range GLCMs.** (a) Variability between the lowest and highest values found in the texture features across all examined exposures calculated form full range GLCMs. (b) Islands of stability in features calculated from full range GLCMs with median exposure as a blue line. *Missing values indicate absence of an island of stability.
(PDF)

**S7 Fig. Effect of restricted range GLCM quantizing.** Exposure dependency of all examined textural features according to different image resolution and reconstruction algorithm binned into a GLCM with 512 grey levels restricted from zero to a maximal intensity of 25 kBq/ml. Measurement points are shown in the first graph, but omitted subsequently. Loess curves are shown without confidence intervals.
(PDF)

**S1 Text. Supporting material and methods.** Full range and restricted range GLCM quantization.
(PDF)

**S2 Text. Supporting results.** Assessing exposure stability of image features when using full range GLCMs.
(PDF)

## Acknowledgments

Thank goes to the staff at the Department of Nuclear Medicine, Inselspital Bern, for assistance with the PET phantom measurements.

## Author Contributions

**Conceptualization:** George Amadeus Prenosil, Thilo Weitzel, Markus Fürstner, Michael Hentschel, Thomas Krause, Bernd Klaeser.

**Data curation:** George Amadeus Prenosil.

**Formal analysis:** George Amadeus Prenosil, Thilo Weitzel.

**Funding acquisition:** Thomas Krause, Axel Rominger, Bernd Klaeser.

**Investigation:** George Amadeus Prenosil.

**Methodology:** George Amadeus Prenosil, Thilo Weitzel.

**Project administration:** George Amadeus Prenosil, Thilo Weitzel, Thomas Krause, Axel Rominger, Bernd Klaeser.

**Resources:** Thomas Krause, Axel Rominger, Bernd Klaeser.

**Software:** George Amadeus Prenosil, Thilo Weitzel.

**Supervision:** George Amadeus Prenosil, Thilo Weitzel, Axel Rominger, Bernd Klaeser.

**Validation:** George Amadeus Prenosil, Thilo Weitzel, Markus Fürstner, Michael Hentschel, Bernd Klaeser.

**Visualization:** George Amadeus Prenosil.

**Writing – original draft:** George Amadeus Prenosil, Thilo Weitzel, Bernd Klaeser.

**Writing – review & editing:** George Amadeus Prenosil, Thilo Weitzel, Markus Fürstner, Michael Hentschel, Thomas Krause, Paul Cumming, Axel Rominger, Bernd Klaeser.

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
