## [Decision Letter · Decision Letter 0]

19 Nov 2019

PONE-D-19-27969

Towards guidelines to harmonize textural features in PET:

Haralick textural features vary with image noise, but exposure-invariant domains enable comparable PET radiomics

PLOS ONE

Dear Dr. Prenosil,

Thank you for submitting your manuscript to PLOS ONE. After careful consideration, we feel that it has merit but does not fully meet PLOS ONE’s publication criteria as it currently stands. Therefore, we invite you to submit a revised version of the manuscript that addresses the points raised during the review process.

We would appreciate receiving your revised manuscript by Jan 03 2020 11:59PM. To enhance the reproducibility of your results, we recommend that if applicable you deposit your laboratory protocols in protocols.io, where a protocol can be assigned its own identifier (DOI) such that it can be cited independently in the future. For instructions see: http://journals.plos.org/plosone/s/submission-guidelines#loc-laboratory-protocols

We look forward to receiving your revised manuscript.

Kind regards,

Mathieu Hatt, MSc, PhD, HDR

Academic Editor

PLOS ONE

Journal Requirements:

Additional Editor Comments (if provided):

Both reviewers are in agreement that the revised paper is much improved compared to the original submission. However, the first one is much more reserved, especially with respect to both the overall rationale and the clinical relevance of the work and estimates the most critical points of his first review were not adequatly adressed. The second reviewer has additional comments that will need to be taken into account in the second revision. However, it will require additional results and analyses in order to adequately respond to all initial and new comments of the first reviewer.

Reviewers' comments:

Reviewer's Responses to Questions

**Comments to the Author**

1. Is the manuscript technically sound, and do the data support the conclusions?

Reviewer #1: Yes

Reviewer #2: Yes

2. Has the statistical analysis been performed appropriately and rigorously? 

Reviewer #1: Yes

Reviewer #2: Yes

3. Have the authors made all data underlying the findings in their manuscript fully available?

Reviewer #1: No

Reviewer #2: Yes

4. Is the manuscript presented in an intelligible fashion and written in standard English?

Reviewer #1: Yes

Reviewer #2: Yes

5. Review Comments to the Author

Reviewer #1: I would like to thank the authors for their answers to my comments. However, their answers do not provide additional rational for the clinical relevance of this study. Although, the study is interesting in its theoretical merit the conclusions drawn are limited. The studied range of exposures is large an authors failed to address its clinical significance. Additionally, the limited set of studied texture features was not addressed. Perhaps other matrices are less prone to the exposure variation.

1. 'The use of a homogenous phantom is completely justified for several reasons, as now stated in the revision:

Line 171 “to quantify a true texture, one must first be able to identify truly homogeneous areas, where any

discerned texture is actually image noise”

We note that any heterogeneous phantom must be scale-invariant if it is to possess the same features over a range of different resolutions, as used in our work (c.f. Lines 137 - 145), and as is required to accommodate multicenter designs.'

Response: I do not agree that the phantom should be scale-invariant. In real life scenario tracer uptake in a tissue is not scale-invariant and thus the control for voxel size in the analysis is important.

2. 'Our result that Haralick feature values vary strongly with exposure, but that there are exposure domains with

stability, is important, and stand in contrast to previous literature. This seems to us more important than attempting

to quantify noise with textural features. In any event, we have incidentally quantified the image noise

thoroughly using nine first-order statistics (Fig. 2 and Table 2). In response to the reviews, we have shifted our

focus to the key point of exposure invariance, and we follow your suggested title change to read,'

Response: here remains the question if the instability is observed for the clinically used exposures, or is the result of wide range used for this study

3. 'Indeed, our main concern here is exposure within a VOI, which is the product of acquisition time and activity

concentration. Therefore, to arrive at comparable feature values between sites, the required exposures will

depend on the sensitivities of the PET/CT systems. These vary over several orders of magnitude depending

on the generation of the PET/CT systems. We used three hour acquisition times as limit to obtain the best

practicably achievable exposure, thus reflecting approaching a gold standard characterized by the smallest

possible noise. Similarly, the three seconds acquisitions represent an according worst case. The extremis

make sure, that clinically relevant exposures with strongly varying acquisition times and strongly varying sensitivity

of the devices fall inside the examined range. '

Response: I understand this reasoning and the differences in the scanner designs. However, could you give an example of a scanner that allows you for clinical image acquisition within 3s per bed position? My concern is that your range, even though it comprises all the clinically relevant times, is so wide that it actually shows the problem mostly in the range which is not clinically used

Reviewer #2: The authors provided a revised manuscript that investigates the dependence of grey level co-occurrence matrix features and several statistical features on exposure in a homogenous Ge-68 phantom. In my opinion, the manuscript improved considerably. Particularly the methods and the results section have been made easier to understand by focussing on one type of discretisation scheme.

I have the following comments:

1. The manuscript is quite long, which may discourage readers from reading it completely. Compare, for example, with Radiology, which allows for 3000 words in the main body, JNM (5000 words, incl. references), EJNMMI (6000 words, incl. references). I found the introduction particularly hard to digest. In the introduction the authors raise multiple points that are nonetheless relevant, but make it difficult for the reader to understand what this study is about. I would recommend to shorten the introduction considerably by summarising some of these points, and, if necessary, integrate some into the discussion.

2. The results contain subsections named “Grey-level co-occurrence matrices” and “Second order statistical features”. I found this somewhat confusing, as second order statistical features are GLCM features (in this work). I would recommend to change these section headers to be more descriptive, or aggregate them into a single section.

3. I would like to commend the authors for focussing on one resampling method only. However, it would be interesting to see what happens when a lower number of bins is selected. For example, Hatt et al. (10.2967/jnumed.114.144055; figure 2) show that some texture feature values (or their correlation with MATV) do depend on the number of bins. The number of bins chosen by the authors is at the high end of this range, which showed relatively few changes. A question that occurred to me was if features extracted from GLCMs with less columns/rows reach stable values earlier? This could be important to inform guidelines. I would recommend 8, 16, 32 bins in addition to the bin numbers already assessed. These values seem to be used more often, see, e.g. Leijenaar et al. (10.1038/srep11075).

4. Lines 103-105: PET/CT sites may be interpreted as anatomical sites instead of centres, which I think is the intended use.

5. Lines 137-139: I am not sure if the reference to nearest neighbourhood operations is necessary. I found it confusing, as it does not seem to bear on the current manuscript.

6. Lines 142-145 “Additionally, the highest …”: This seems to be more fitting for the discussion.

7. Lines 158-160 “The resulting exposures …”: This seems to be more fitting for the discussion. Moreover, what exposures are clinically relevant?

8. Lines 171-173 “(to quantify …”. This seems to be more fitting to the introduction.

9. Line 202: What is a multi-paradigm software?

10. Lines 217-219 “The large number …”: I did not understand the justification.

6. PLOS authors have the option to publish the peer review history of their article (what does this mean?). If published, this will include your full peer review and any attached files.

Reviewer #1: No

Reviewer #2: No

---

## [Author Response · Author response to Decision Letter 0]

16 Dec 2019

Journal Requirements:

Author: We followed the style template

Author: The data previously not shown was placed in the supporting information section in S1 Fig

Additional Editor Comments (if provided):

Both reviewers are in agreement that the revised paper is much improved compared to the original submission. However, the first one is much more reserved, especially with respect to both the overall rationale and the clinical relevance of the work and estimates the most critical points of his first review were not adequatly adressed. The second reviewer has additional comments that will need to be taken into account in the second revision. However, it will require additional results and analyses in order to adequately respond to all initial and new comments of the first reviewer.

Reviewers' comments:

Reviewer's Responses to Questions

Comments to the Author

1. Is the manuscript technically sound, and do the data support the conclusions?

Reviewer #1: Yes

Reviewer #2: Yes

2. Has the statistical analysis been performed appropriately and rigorously? 

Reviewer #1: Yes

Reviewer #2: Yes

3. Have the authors made all data underlying the findings in their manuscript fully available?

Reviewer #1: No

Author: We added the tag DICOM Files to the file archive (.zip) 20180717 at https://osf.io/36c7s/. Additionally, we uploaded tsv-files (3bit_8bit_Reports.zip) with all the extracted noise and feature values from the DICOM files, analyzed with 3 to 8 bit GLCMS. We added a section “Disclosure”, were the availability of the data is mentioned together with the download link (Line 572).

Reviewer #2: Yes

4. Is the manuscript presented in an intelligible fashion and written in standard English?

Reviewer #1: Yes

Reviewer #2: Yes

5. Review Comments to the Author

Reviewer #1: I would like to thank the authors for their answers to my comments. However, their answers do not provide additional rational for the clinical relevance of this study. Although, the study is interesting in its theoretical merit the conclusions drawn are limited. The studied range of exposures is large an authors failed to address its clinical significance. Additionally, the limited set of studied texture features was not addressed. Perhaps other matrices are less prone to the exposure variation.

Author: Clinical relevance of this study is attested based on two important cited works. One work focused on Haralick features from 6-bit GLCMs measured with a PET/CT similar to our own, and the other study pro-vides time activity curves with exposures in the ranges examined in our work (Lines 427 to 449). To make matters clearer, we have provided easy-to-understand calculations of exposure in this new text. As men-tioned in our letter to the editor, we analyzed in-depth a particular set of features, the widely employed Haralick features. Extending this study to include the countless possible feature sets is practically untena-ble, if we were to apply the same degree of explanatory power. However, we do state at the end of discus-sion that measurements of other feature classes should be done.

The abstract mentions now the possibility of modern PET/CT systems to attain the stable exposures (Lines 56 and 57).

1. 'The use of a homogenous phantom is completely justified for several reasons, as now stated in the revision:

Line 171 “to quantify a true texture, one must first be able to identify truly homogeneous areas, where any

discerned texture is actually image noise”

We note that any heterogeneous phantom must be scale-invariant if it is to possess the same features over a range of different resolutions, as used in our work (c.f. Lines 137 - 145), and as is required to accommodate multicenter designs.'

Response: I do not agree that the phantom should be scale-invariant. In real life scenario tracer uptake in a tissue is not scale-invariant and thus the control for voxel size in the analysis is important.

Author: We have to disagree; exactly because of the scale-variance of differently resolved PET/CT acquisitions, the procedure used to measure scale sensitive feature must itself be scale invariant. We have tried to clarify this key issue in lines 507 to 516. But we agree, that image resolution must be adapted to the desired clinical question. This is why we provide results for three commonly used image resolutions.

One should not confuse image resolution with voxel size. The resolution of a PET/CT system is given by its systemic PSF (MTF) convolved with the chosen post-reconstruction filter. The voxel size must just follow accordingly, i.e. to the Nyquist-Shannon sampling theorem. For this reason, the problematic of image reso-lution and texture has been already addressed at lines in the previous submission (Now lines 536 to 542). We extended this paragraph at line 539 and 540.

2. 'Our result that Haralick feature values vary strongly with exposure, but that there are exposure domains with

stability, is important, and stand in contrast to previous literature. This seems to us more important than attempting

to quantify noise with textural features. In any event, we have incidentally quantified the image noise

thoroughly using nine first-order statistics (Fig. 2 and Table 2). In response to the reviews, we have shifted our

focus to the key point of exposure invariance, and we follow your suggested title change to read,'

Response: here remains the question if the instability is observed for the clinically used exposures, or is the result of wide range used for this study

Author: The newly cited works (Lines 428 to 450) have exposures that clearly encompass the range of exposures used for our analysis. Furthermore, instability over a wide range of exposures and is seen in the slope of the feature value-curves. That’s why we are also providing the first deviation in S4 Fig to show this fact. Furthermore, it is important to avoid exposure regimes that lead to unstable features (Line 782 and 783). Our work surely shows the extreme but not the impossible (c.f. Lines 428 to 450).

3. 'Indeed, our main concern here is exposure within a VOI, which is the product of acquisition time and activity

concentration. Therefore, to arrive at comparable feature values between sites, the required exposures will

depend on the sensitivities of the PET/CT systems. These vary over several orders of magnitude depending

on the generation of the PET/CT systems. We used three hour acquisition times as limit to obtain the best

practicably achievable exposure, thus reflecting approaching a gold standard characterized by the smallest

possible noise. Similarly, the three seconds acquisitions represent an according worst case. The extremis

make sure, that clinically relevant exposures with strongly varying acquisition times and strongly varying sensitivity

of the devices fall inside the examined range. '

Response: I understand this reasoning and the differences in the scanner designs. However, could you give an example of a scanner that allows you for clinical image acquisition within 3s per bed position? My concern is that your range, even though it comprises all the clinically relevant times, is so wide that it actually shows the problem mostly in the range which is not clinically used

Author: It seems that Reviewer #1 has not correctly understood the concept of exposure, as he/she repeatedly re-fers to acquisition time, which is not germane in this context. On an older PET/CT system with ten-fold lower sensitivity, a 30 s would correspond to our 3 s. On the other hand, with the promise of roughly 14 times better sensitivity, a Siemens Biograph Vision can obtain stable VAR values in about 3.7 min instead of 52 min, as stated in our manuscript. As we have mentioned, acquisitions times have no meaning per se if used without the expected activity concentration and the without the characterization of the corresponding PET/CT system. Furthermore, the issue is addressed with the citation and discussion of the two example studies (Lines 428 to 450).

Reviewer #2: The authors provided a revised manuscript that investigates the dependence of grey level co-occurrence matrix features and several statistical features on exposure in a homogenous Ge-68 phantom. In my opinion, the manuscript improved considerably. Particularly the methods and the results section have been made easier to understand by focussing on one type of discretisation scheme.

I have the following comments:

1. The manuscript is quite long, which may discourage readers from reading it completely. Compare, for example, with Radiology, which allows for 3000 words in the main body, JNM (5000 words, incl. references), EJNMMI (6000 words, incl. references). I found the introduction particularly hard to digest. In the introduction the authors raise multiple points that are nonetheless relevant, but make it difficult for the reader to understand what this study is about. I would recommend to shorten the introduction considerably by summarising some of these points, and, if necessary, integrate some into the discussion.

Author: Yes, we agree that this manuscript is wordy, in part due to the complexity of the issues arising in it, and in part due to earlier requests for further elaboration upon review. Nonetheless, we have substantially shorted the introduction by removing the EARL section and some other parts (shown with markups) and by moving other text to the revised “Discussion”. The concept of exposure is now largely presented in the revised “Ma-terial and Methods” section, but, for the sake of brevity, is not described in detail in the Introduction. Former Figure 8 was moved to the “Supporting Information” section to become S7 Fig.

2. The results contain subsections named “Grey-level co-occurrence matrices” and “Second order statistical features”. I found this somewhat confusing, as second order statistical features are GLCM features (in this work). I would recommend to change these section headers to be more descriptive, or aggregate them into a single section.

Author: Agreed. Second order features as well as GLCFs are now consistently called Haralick features. Additionally, we have changed the results section from “First order statistical features” to “Exposure dependency of first order statistical features”. The section “grey level co-occurrence matrices” in “Results” has been changed to read “Exposure dependency of grey level co-occurrence matrices”, and the “Results” section “Second order statistical features” is now entitled “Exposure dependency of Haralick features”.

3. I would like to commend the authors for focussing on one resampling method only. However, it would be interesting to see what happens when a lower number of bins is selected. For example, Hatt et al. (10.2967/jnumed.114.144055; figure 2) show that some texture feature values (or their correlation with MATV) do depend on the number of bins. The number of bins chosen by the authors is at the high end of this range, which showed relatively few changes. A question that occurred to me was if features extracted from GLCMs with less columns/rows reach stable values earlier? This could be important to inform guidelines. I would recommend 8, 16, 32 bins in addition to the bin numbers already assessed. These values seem to be used more often, see, e.g. Leijenaar et al. (10.1038/srep11075).

Author: The second resampling method, which was suggested in the earlier review of this manuscript, was retained in the supporting material, as it is brief and potentially helpful future studies based on present results Never-theless, former “Figure 8” was moved to the “Supporting Information” section to become “S7 Fig”. Following the reviewer’s suggestion, we ran the analysis using 8, 16, and 32 bins (Line 218), and have extended the “Results” section accordingly (Fig. 6 and Fig. 7, Lines 347 to 349), as well as the “Discussion” (Lines 461 and 462). The exercise further documented the relative invariance of Haralick features regarding bin size. Hatt et al. (10.2967/jnumed.114.144055) was introduced at line 349. Leijenaar et al. (10.1038/srep11075) be-came useful at line 386.

4. Lines 103-105: PET/CT sites may be interpreted as anatomical sites instead of centres, which I think is the intended use.

Author: Yes, we now use the more correct term PET/CT systems (Line 107)

5. Lines 137-139: I am not sure if the reference to nearest neighbourhood operations is necessary. I found it confusing, as it does not seem to bear on the current manuscript.

Author: Yes, this is important, as it is exactly this type of features that arise from neighborhood operations. Other definitions use other operations and might have other exposure dependencies (c.f. outlook). Furthermore, this section was already introduced in response to concerns raised in the previous reviewer. This section was moved to the Discussion (Lines 507 - 516).

6. Lines 142-145 “Additionally, the highest …”: This seems to be more fitting for the discussion.

Author: Yes, c.f. above. Was moved to the discussion.

7. Lines 158-160 “The resulting exposures …”: This seems to be more fitting for the discussion. Moreover, what exposures are clinically relevant?

Author: Yes. We have deleted most of this text, but added some new information specifying the clinical signifi-cance in the revised Discussion (Lines 428 to 450). The clinically relevant exposures were added to Fig 5, with data taken from the first example (Ref. [47]).

The abstract mentions now the possibility of modern PET/CT systems to attain the stable exposures (Lines 56 and 57).

8. Lines 171-173 “(to quantify …”. This seems to be more fitting to the introduction.

Author: We guess the reviewer means “Discussion”. Moved to discussion (Lines 493 and 494) to shorten the introduction.

9. Line 202: What is a multi-paradigm software?

Author: We have added a reference explaining said principles of software development (Line 214).

10. Lines 217-219 “The large number …”: I did not understand the justification.

Author: The number of grey levels (information content) of a GLCM must be matched by a sufficiently high number of measurement points (voxels). Otherwise, the GLCM will have void entries were none should exist. A sta-tistical analysis of such GLCMs then becomes unreliable. This caveat is now mentioned in the text (Lines 232 and 233).

6. PLOS authors have the option to publish the peer review history of their article (what does this mean?). If published, this will include your full peer review and any attached files.

Do you want your identity to be public for this peer review? For information about this choice, including consent withdrawal, please see our Privacy Policy.

Reviewer #1: No

Reviewer #2: No

---

## [Decision Letter · Decision Letter 1]

11 Feb 2020

Towards guidelines to harmonize textural features in PET:

Haralick textural features vary with image noise, but exposure-invariant domains enable comparable PET radiomics

PONE-D-19-27969R1

Dear Dr. Prenosil,

We are pleased to inform you that your manuscript has been judged scientifically suitable for publication and will be formally accepted for publication once it complies with all outstanding technical requirements.

With kind regards,

Mathieu Hatt, MSc, PhD, HDR

Academic Editor

PLOS ONE

Additional Editor Comments (optional):

Both reviewers are now satisfied with the revised manuscript. Please take into account the few remaining remarks of reviewer #1 while preparing the final version.

Reviewers' comments:

Reviewer's Responses to Questions

**Comments to the Author**

1. If the authors have adequately addressed your comments raised in a previous round of review and you feel that this manuscript is now acceptable for publication, you may indicate that here to bypass the “Comments to the Author” section, enter your conflict of interest statement in the “Confidential to Editor” section, and submit your "Accept" recommendation.

Reviewer #1: All comments have been addressed

Reviewer #2: (No Response)

2. Is the manuscript technically sound, and do the data support the conclusions?

Reviewer #1: Yes

Reviewer #2: Yes

3. Has the statistical analysis been performed appropriately and rigorously? 

Reviewer #1: Yes

Reviewer #2: Yes

4. Have the authors made all data underlying the findings in their manuscript fully available?

Reviewer #1: Yes

Reviewer #2: Yes

5. Is the manuscript presented in an intelligible fashion and written in standard English?

Reviewer #1: Yes

Reviewer #2: Yes

6. Review Comments to the Author

Reviewer #1: (No Response)

Reviewer #2: The authors have adequately addressed my previous remarks.

The following comments are extremely minor:

1. The authors should not that not all references were parsed correctly, e.g. lines 100, 366, 422, 427.

2. Line 221-222: It would be better to explicitly define x-direction and y-direction in the context of a GLCM matrix, to avoid confusion with directions in the image grid.

3. Line 468-469: The sentence structure is not correct, particularly the “… extrema and, and as we have shown previously [47], and …” part.

4. Line 509: I would prefer d > 1 instead of d >> 1.

7. PLOS authors have the option to publish the peer review history of their article (what does this mean?). If published, this will include your full peer review and any attached files.

Reviewer #1: No

Reviewer #2: Yes: Alex Zwanenburg

---

## [Editor Report · Acceptance letter]

20 Feb 2020

PONE-D-19-27969R1 

Towards guidelines to harmonize textural features in PET:
Haralick textural features vary with image noise, but exposure-invariant domains enable comparable PET radiomics 

Dear Dr. Prenosil:

I am pleased to inform you that your manuscript has been deemed suitable for publication in PLOS ONE. Congratulations! Your manuscript is now with our production department. 

With kind regards,

on behalf of

Dr. Mathieu Hatt 

Academic Editor

PLOS ONE